# Characteristics of the First Domestic Duck-Origin H12N8 Avian Influenza Virus in China

**DOI:** 10.3390/ijms26062740

**Published:** 2025-03-18

**Authors:** Conghui Zhao, Jiacheng Huang, Chunping Zhang, Yang Wang, Xiaoxuan Zhang, Sha Liu, Haoxi Qiang, Huanhuan Wang, Hangyu Zheng, Mingzhi Zhuang, Yanni Peng, Fuzai Chen, Xiancheng Zeng, Ji-Long Chen, Shujie Ma

**Affiliations:** Fujian Province Joint Laboratory of Animal Pathogen Prevention and Control of the “Belt and Road”, College of Animal Sciences, Fujian Agriculture and Forestry University, Fuzhou 350002, China; zhaoconghui19@163.com (C.Z.); m13395087570@163.com (J.H.); m18806027576@163.com (C.Z.); wangyangvv2023@163.com (Y.W.); zhangxiaoxuan5426@163.com (X.Z.); m15037362321@163.com (S.L.); siriusxx_010106@163.com (H.Q.); wanghuan12282487@163.com (H.W.); m17350190341@163.com (H.Z.); m19835211797@163.com (M.Z.); pengpenny39@163.com (Y.P.); chenfuzai0523@163.com (F.C.); zxc206@163.com (X.Z.); chenjilong@fafu.edu.cn (J.-L.C.)

**Keywords:** avian influenza virus, H12N8, evolution, pathogenicity, transmissibility

## Abstract

The H12 subtypes of avian influenza viruses (AIVs) are globally prevalent in wild birds, occasionally spilling over into poultry. In this study, we isolated an H12N8 virus from ducks in a live poultry market. Full genomic analysis revealed that the virus bears a single basic amino acid in the cleavage site of the hemagglutinin gene. Phylogenetic analysis revealed that the eight gene segments of the H12N8 virus belong to the Eurasian lineage and the HA gene was clustered with wild bird-originated H12 viruses, with its NP gene showing the highest nucleotide similarity to 2013-like H7N9 viruses. The H12N8 virus replicated effectively in both mammalian and avian cells without prior adaptation. Moreover, the H12N8 virus could infect and replicate in the upper respiratory tract of BALB/c mice without prior adaptation. The H12N8 virus replicated and transmitted inefficiently in both ducks and chickens and hardly triggered high hemagglutination inhibition (HI) antibody titers in the inoculated and contact animals. These results suggest that the wild bird-origin H12N8 virus has reassorted with viruses circulating in domestic poultry, but it inefficiently replicates and transmits in avian hosts. Our findings demonstrate that H12N8 AIV has emerged in domestic poultry, emphasizing the importance of active surveillance of AIVs in both wild and domestic birds.

## 1. Introduction

Avian influenza virus (AIV) belongs to the *Orthomyxoviridae* family, with its genome consisting of eight negative-sense single-stranded RNAs. The viral genome mainly encodes the basic polymerase 2 (PB2), basic polymerase 1 (PB1), acidic polymerase (PA), hemagglutinin (HA), nucleoprotein (NP), neuraminidase (NA), matrix 1 (M1), matrix 2 (M2), nonstructural protein 1 (NS1), and nonstructural protein 2 (NS2). Until now, 16 distinct HA and 9 different NA subtypes of influenza viruses have been detected in avian species, while H17N10 and H18N11 are only identified in bats [1,2]. Furthermore, the H19 strain of the virus has been identified in fecal samples collected from wild avian species [3,4].

Wild aquatic birds are the primary reservoirs of AIVs. AIVs can be readily transmitted or undergo genetic reassortment between wild birds and domestic waterfowl when these birds share the same water area. Ducks and geese are often raised in open environments that lack effective biosecurity measures. The highly pathogenic AIVs (HPAIVs) of H5 and H7 have caused several outbreaks in poultry and wild birds in many countries in recent years [5,6,7,8,9,10,11,12,13,14,15]. Since AIVs of diverse subtypes asymptomatically infect and replicate in domestic ducks, vaccination programs in these hosts are rarely implemented. Thus, ducks seem to be an important host for the recombination of AIVs from wild birds and domestic poultry. Most importantly, some subtypes of AIVs, including H3N8, H7N9, and H10N8, with their genes derived from duck-origin viruses, have acquired the ability to infect humans [16,17,18].

The H12 subtype virus was identified in the United States as early as 1975 with the virus named A/gadwall/Wisconsin/13/1975 (H12N5) (GenBank number: CY179859.1) according to a retrospective study. Although the H12 subtype viruses were infrequently detected in wild birds and domestic poultry, these H12 viruses exhibited a high degree of genetic reassortment with other subtype viruses, serving as an important gene pool of AIVs [19,20,21,22,23]. The H12N2 virus isolated from Anas falcata in Dongting Lake wetland showed gene segments derived from multiple hosts, including ducks, geese, and wild birds, indicating reassortment of the virus between wild birds and domestic poultry [19]. Intercontinental reassortment of gene segments has been observed in Eurasian and North American H12N1, H12N2, and H12N3 viruses, suggesting the transhemispheric spread of these viruses [20,22].

The H12N8 virus was first isolated from wild bird feces collected in Hunan East Dongting Lake National Nature Reserve in 2011 in China [24]. Researchers found that the wild bird-origin H12N8 was a novel reassortment of different subtype AIVs circulating around the Eurasian continent. This wild bird-origin H12N8 virus was classified as a low pathogenic AIV (LPAIV) based on the monobasic cleavage site motif in its HA protein. However, the biological characteristics of this wild bird-origin H12N8 virus remain largely unknown.

In this study, we report the first isolation of H12N8 virus from ducks in a live poultry market in Fujian Province in China. Previous studies have demonstrated that the spillover of some subtypes of AIVs including H3N8, H9N2, H5N1, H7N7, and H7N9 from wild birds to domestic poultry may result in the widespread of the virus or generate reassorted viruses in poultry [25,26,27]. Thus, it is important to assess the biological characteristics of the first H12N8 virus isolated from ducks although the virus is classified as a LPAIV. To this end, we performed phylogenetic analysis, growth kinetics in mammalian and avian cells, pathogenicity in mice, and transmission in ducks and chickens to understand the potential threat of the H12N8 virus to public health and the poultry industry. These findings help us better understand the properties of the H12N8 virus and highlight the need for continuous surveillance of AIVs circulating in domestic birds.

## 2. Results

### 2.1. Global Prevalence of H12 Subtype Viruses

To fully understand the global prevalence of H12 subtype AIVs, all the HA sequences of H12 AIVs were retrieved from both GISAID and GenBank databases between January 1975 and February 2025. The results showed that about 61.8% (328/531) of the H12 viruses were H12N5 subtype (Figure 1A), while only 18 strains (3.4%, 18/531) of H12N8 were identified in wild birds including ruddy turnstone, mallard, northern pintail, common teal, wild duck, and ring-necked duck (Appendix A). The H12 subtype AIVs were mainly distributed among wild birds (95.2%), with 60.6% of the viruses identified in ruddy turnstone and mallard, while only 43 H12 viruses including H12N1, H12N2, H12N5, H12N6, H12N7, and H12N8 were detected in ducks (Appendix A). The geographical distribution results showed that the H12 subtype viruses were mainly detected in the United States (Figure 1C). In China, the H12 viruses, including the H12N8 isolate in this study, were only detected in Shanghai, Taiwan, Hunan, Shandong, Shanxi, Jiangxi, and Fujian, all of which are located along the East Asia–Australasia migratory flyway (Figure 1D).

### 2.2. Isolation and Molecular Characteristics of a Duck-Origin H12N8 Virus

To determine the viral evolution in domestic birds, we collected 1190 samples from live poultry markets in chickens, ducks, and pigeons in Fujian Province between November 2019 and December 2023. We collected 25, 209, 112, 131, and 713 samples from 2019 to 2023 annually. The samples were inoculated into 10-day-old embryonated chicken eggs for virus isolation. Subsequently, the positive sample was amplified in Specific pathogen free (SPF) chicken embryo eggs, and their HA and NA subtype were identified using Sanger sequencing. Virus isolation yielded 116 AIVs, with an annual distribution as follows: 26 isolates in 2020, 17 in 2021, 5 in 2022, and 68 in 2023. The subtype distribution of these viruses is H3 (30.17%, 35/116), H6 (27.59%, 32/116), H4 (16.38%, 19/116), H9 (12.93%, 15/116), H10 (5.17%, 6/116), H1 (4.31%, 5/116), H11 (2.59%, 3/116), and H12 (0.86%, 1/116). We detected only one H12N8 subtype virus in these samples and designated it as A/duck/Fujian/D62/2020 (H12N8) (DK/FJ/D62/2020). We found that the DK/FJ/D62/2020, which was detected in this study, was the first H12N8 isolate in poultry in China after a thorough review of the literature and databases. The amino acid residues at the cleavage site of the HA protein are PQVQNR↓GLF, representing low pathogenicity in chickens (Table 1). The amino acids at the receptor binding site in the HA protein are Q226 and G228 (H3 numbering), which indicates that this virus may preferentially bind to the avian-type receptor (α-2,3 sialic acid) (Table 1). Most of the substitutions for mammalian adaptation were not found in PB2, NP, NA, or M2 (Table 1), while several mammalian adaptive mutations in PB1, PA, NP, M1, and NS1 were presented in the DK/FJ/D62/2020 virus (Table 1). As shown in Appendix A, the potential glycosylation site motifs N-X-S/T revealed nine sites at positions 27, 28, 140, 151, 222, 302, 309, 496, and 523 in the HA protein and six sites at positions 46, 54, 84, 144, 293, and 398 in the NA protein.

### 2.3. Phylogenetic Analysis of the H12N8 Virus

To systematically understand the evolution of the H12N8 virus, we made a phylogenetic analysis of the eight genes. A BLAST search in the GISAID and GenBank databases indicated a high nucleotide identity of the DK/FJ/D62/2020 virus with H12, H1, H11, H3, and H7 subtype viruses isolated in ducks, environment, and humans in Asian countries. The HA gene exhibited relatively low similarity (94.93%) to the duck H12N5 virus isolated in Vietnam, while the other seven genes shared higher identity with other viruses (97.38–99.25%) (Appendix A). All eight gene segments of the DK/FJ/D62/2020 virus belong to the Eurasian lineage and are markedly distinct from the North American avian influenza A virus lineage (Figure 2 and Figure 3). To better understand the genesis and evolutionary timeline, we constructed a Bayesian time-resolved tree with HA of the H12N8 virus and the representative viruses of different H12 subtypes detected in Eurasian and North American countries since 1975. We found that the HA gene of the H12N8 virus was closest to the isolate of A/duck/Vietnam/G18/2009 (H12N5), which was clustered with wild bird-originated H12 viruses (Figure 2A). The phylogenetic trees of the rest seven genes of the H12N8 virus were generated by using the maximum-likelihood method with the IQ-TREE algorithm. The NA gene was clustered with the human H10N8 and H3N8 viruses, while distinct from the H5N8 branch (Figure 2B). The PB2 and M genes were closest to A/environment/Fujian/EV01/2020 (H11N3) with 99.25% and 99.08% identities, respectively. The PB1 and PA genes were closest to the H3N2 and H1N2 viruses, respectively, which were clustered with the domestic poultry virus branches. Most importantly, the NP gene shares the highest nucleotide similarity with a 2013-like H7N9 virus isolate A/Fujian/33845/2017 (H7N9) in humans (Figure 3, Appendix A). The NS gene shared the highest similarity with another branch of the H7N9 virus (distinct from the 2013-like H7N9) A/duck/Zhejiang/S4489/2014 (H7N9), with an identity of 99.20%. These results implied that the DK/FJ/D62/2020 is a novel reassorted H12N8 virus through the complex recombination of multiple avian and human influenza viruses.

### 2.4. Replication of the H12N8 Virus in the Mammalian and Avian Cells

To evaluate the replication characteristics of the H12N8 virus in avian and mammalian cells, multicycle growth curves of the DK/FJ/D62/2020 virus were assessed in UMNSAH/DF1 (DF1), Madin-Darby canine kidney (MDCK), and human lung adenocarcinoma epithelial (A549) cells. Briefly, the cells were grown on 12-well plates and infected with the virus at an MOI of 0.01. The supernatant was collected at indicated times and titrated in MDCK cells. The titers of the DK/FJ/D62/2020 virus on DF1, MDCK, and A549 cells ranged from 3.25 to 4.75 log_10_ TCID_50_/mL (Figure 4A), 3.25–6.50 log_10_ TCID_50_/mL (Figure 4B), and 1.25–2.25 log_10_ TCID_50_/mL (Figure 4C) at 12 h, 24 h, 36 h, and 48 h p.i., respectively. Notably, the virus titers on MDCK cells were relatively higher than those in DF1 and A549 cells at 12 h, 24 h, 36 h, and 48 h p.i. (Figure 4). These results demonstrate that the H12N8 virus can replicate in both avian and mammalian cells without prior adaptation.

### 2.5. Replication and Pathogenicity of the H12N8 Virus in Mice

To evaluate the pathogenicity of the H12N8 virus to mammals, we tested the replication and virulence in BALB/c mice by intranasal inoculation with the DK/FJ/D62/2020 virus at a dose of 10^6^ 50% egg infectious doses (EID_50_) and monitored weight loss for 14 d p.i. Viral titers in nasal turbinates, lungs, brains, spleens, and kidneys were determined on day 3 d p.i. with embryonated chicken eggs. The results demonstrated that the H12N8 virus induced a modest reduction in body weight among mice inoculated with the DK/FJ/D62/2020 strain, as opposed to the Phosphate-Buffered Saline (PBS) control group (Figure 5A). The DK/FJ/D62/2020 virus replicated in the turbinates of the mice with the virus titers ranging from 1.75 to 2.25 log_10_ EID_50_/mL. Virus was not detected in the lungs, brains, spleens, or kidneys (Figure 5B). These results suggested that the H12N8 virus could infect and obtain the replication ability in the upper respiratory tract of mice without prior adaptation. Although viruses were not recovered in the lungs of infected mice, pulmonary pathological lesions were detected in the lungs of the infected mice compared with the control group by using hematoxylin and eosin (H&E) staining. The structure of the small bronchi and alveoli of the control group mice was intact (Figure 5C), while the infected group mice showed minor damage within the bronchioles without inflammatory cell infiltration, leading to epithelial cell detachment (Figure 5D).

### 2.6. Replication and Transmission of H12N8 Virus in Ducks

The DK/FJ/D62/2020 virus was the first isolate in poultry in China. The replication and transmission in ducks are poorly understood. To investigate the replication of the virus in ducks, three SPF ducks were intranasally inoculated with the DK/FJ/D62/2020 virus at a dose of 10^6^ EID_50_. The birds were euthanized on 3 d p.i. for virus titration in different organs. The virus was only recovered from the larynx of two out of three ducks, indicating that the DK/FJ/D62/2020 virus could infect ducks but replicated poorly in ducks’ organs (Figure 6A). All three ducks showed no hemorrhage at the laryngeal inlet (Figure 6B). In the transmission study, viruses were detected in the oropharyngeal swabs on day 4 and day 3 from both inoculated and contact ducks, respectively, while the virus in the cloacal swabs was only detected from one inoculated duck at day 8 (Figure 6C). Interestingly, although the virus was able to replicate in the inoculated and contact ducks at low levels in the upper respiratory tract, it did not trigger the production of high hemagglutination inhibition (HI) antibody titers in the inoculated and contact ducks (Figure 6D). These findings imply that ducks may serve as asymptomatic carriers of the H12N8 virus, indicating a potential for silent limited transmission within avian populations.

### 2.7. Replication and Transmission of the H12N8 Virus in Chickens

To investigate the virus replication and transmission in chickens, three SPF chickens were intranasally inoculated with the DK/FJ/D62/2020 virus with 10^6^ EID_50_. The birds were euthanized on 3 d p.i. to detect virus titers in different organs. The virus was recovered from the larynx and trachea of all three chickens, indicating that the DK/FJ/D62/2020 virus could infect the birds and replicate in the upper respiratory tract (Figure 7A). Importantly, all three chickens showed hemorrhage at the laryngeal inlet (Figure 7B). In the transmission study, viruses were detected in the oropharyngeal swabs on day 2 and day 4 in all three inoculated chickens, while it was detected in only one bird at day 3 in the contact group (Figure 7C). The virus triggered production of low HI antibody titers in the inoculated chickens but not in contact chickens (Figure 7D). These findings suggest that chickens are relatively susceptible hosts for the H12N8 virus, which can replicate in the upper respiratory tract but is not readily transmitted via direct contact.

## 3. Discussion

AIVs exhibit a broad geographic distribution and are prevalent in a variety of hosts, ranging from domestic poultry and wild birds to mammals [17,28,29]. While H12 subtype AIVs are mainly circulating in wild birds globally and rarely detected in domestic poultry [19,20,21,23,24], we reported the first isolation of an H12N8 virus in ducks temporarily raised in live poultry markets of southeastern China. Genomic analysis showed that the H12N8 isolate was a reassortment of viruses that originated from wild and domestic waterfowl, and the NP gene shared a high degree of similarity with 2013-like H7N9 HPAIVs.

The H12 subtype influenza viruses are mainly circulating in wild waterfowl, with only 6% of the H12 viruses isolated from ducks. The HA gene of DK/FJ/D62/2020 is closely related to H12 viruses of wild birds and showed the highest similarity with an H12N5 duck isolate virus, indicating that the H12 subtype viruses of wild bird origin are gradually entering poultry populations. When these viruses replicate in poultry, they are likely to become more transmittable or even become highly pathogenic to chickens by undergoing extensive reassortments and/or acquiring mutations in some genes [10,11]. For example, H7N9 viruses, which carried H7 and N9 genes from wild birds and internal genes from chicken H9N2 viruses, emerged in 2013 and have since caused huge economic losses and over 1560 human infections in China [10,11,18,30,31,32]. After four years of epidemic spread, these H7N9 viruses acquired four basic amino acid insertions at the cleavage site of the HA protein and became highly pathogenic to chickens in 2017, causing huge damage to public health and the poultry industry [10,11,33,34]. The NP gene of DK/FJ/D62/2020 in this study was closely related to H7N9 HPAIVs in Fujian province, likely due to the co-circulating and recombination of these viruses in local live poultry markets. Although the implementation of bivalent H5/H7 inactivated vaccine in China has significantly reduced the H7N9 virus prevalence in chickens, the virus has not been eliminated in China. Thus, continuous surveillance of the AIVs circulating in wild birds and domestic poultry is crucial for the control and prevention of the diseases.

Domestic ducks and wild birds often share the same water areas in the breeding process. Since many AIVs, including H5 and H7 HPAIVs, replicate asymptomatically in ducks, the farmers lack motivation to vaccinate their flocks. Thus, the control and prevention of AIVs in waterfowl continue to be a significant and persistent concern. The wide distribution of LPAIVs, especially the H3, H4, H6, H9, and H10 subtypes, makes them prone to reassort with wild bird origin viruses. The emerged H5N8, H5N6, and H5N1 viruses result from complex reassortment with local viruses such as H3N2, H4N6, H6N6, and H9N2, producing multiple virus genotypes [7,8,35]. The recent reports that Clade 2.3.4.4b H5N1 HPAIVs detected in dairy cattle and cats indicate that the risk of cross-species transmission should not be overlooked, since ingestion of feed contaminated with feces from HPAIV-infected wild birds is presumed to be the most likely initial source of infection on dairy farms [36,37,38,39,40,41,42]. The isolation of the H12N8 virus in this study reveals that wild bird-origin viruses are susceptible to genetic reassortment with other prevalent AIVs in poultry. It is not surprising that the H12N8 virus does not replicate or transmit efficiently in both birds and mice, as many AIVs originating from wild birds may require adaptation in domestic poultry or mammalian animals. Therefore, the active surveillance of AIVs in both domestic poultry and wild birds should be strengthened to detect potentially threatening viruses.

The mouse model is widely used in the evaluation of influenza virus pathogenicity [10,30,43,44,45,46]. Upon infection with AIVs, mice display a range of symptoms, from mild to severe anorexia, huddling for warmth, hunching as a sign of discomfort, and fur ruffling indicative of distress [47]. Certain strains of H5 and H7 HPAIVs are known to be lethal in mice, exhibiting robust replication within the respiratory tract or even a systemic infection, with viruses detected in the spleen, kidney, and brain [8,11,48]. The H12N8 virus caused only slight body weight loss and was restricted to the upper respiratory tract, showing limited pathogenicity in this mammalian host. However, the detection of pulmonary pathological lesions without virus recovery in the lungs suggests that the virus has induced host responses, leading to tissue damage, even in the absence of high viral loads.

We have isolated only one H12N8 virus from ducks in this study. This could be due to two main reasons. Firstly, the sample size is relatively small, and since the samples were collected only in Fujian Province, they may not reflect the full scope of the virus’s distribution. Secondly, as our study demonstrates, the H12N8 virus does not replicate and transmit efficiently in ducks and chickens, potentially restricting its effective spread among birds.

In summary, we describe the characteristics of the first H12N8 isolate from domestic ducks in China. The H12N8 virus studied here exhibits low pathogenicity to both avian and mammalian hosts, which is unlikely to raise significant public concern. However, the epidemic risks posed by the co-circulation of AIVs among wild birds and domestic poultry persist. Our findings highlight the critical need for continuous surveillance of AIVs in wild and domestic birds, given their role as reservoirs for emerging strains with potential zoonotic and pandemic threats.

## 4. Materials and Methods

### 4.1. Ethics Statements and Facility

The samples and live virus experiments were conducted in the enhanced biosafety level 2 (BSL2+) facility in Fujian Agriculture and Forestry University. The animal studies were carried out in strict compliance with the recommendations of the Guide for the Care and Use of Laboratory Animals of the Ministry of Science and Technology of China. The protocol for animal studies was approved by the Committee on the Ethics of Animal Experiments at Fujian Agriculture and Forestry University (PZCASFAFU21001).

### 4.2. Cells, Eggs, and Animals

MDCK, A549 cells, and DF1 cells were purchased from the American Type Culture Collection (ATCC, Manassas, VA, USA). The cells were grown in Dulbecco’s modified Eagle’s medium (DMEM) supplemented with 10% fetal bovine serum and antibiotics at 37 °C in 5% CO_2_. SPF-embryonated chicken and duck eggs were obtained from Jinan SPAFAS Poultry Co., Ltd. Six-week-old female BALB/c mice were purchased from Zolgene Biotechnology Co., Ltd., Fuzhou, China.

### 4.3. Sample Collection and Virus Isolation

Cloacal and tracheal swab samples of birds including chickens, ducks, and pigeons were collected from live poultry markets in Fuzhou, Fujian province, between November 2019 and December 2023. Each sample was placed in 2 mL of PBS supplemented with penicillin (2000 U/mL) and streptomycin (2000 U/mL). The samples were maintained in 2–8 °C and shipped to the laboratory in sealed containers. All the samples were centrifuged at 4 °C at 3000× *g* for 10 min, and 200 μL of the supernatants were inoculated into 10-day-old embryonated chicken eggs for 48 h at 37 °C. The allantoic fluid was collected and tested for HA activity with 1% chicken red blood cells. Virus positive samples for stock preparation were propagated in 10-day-old SPF-embryonated chicken eggs and stored at −70 °C before use.

### 4.4. Sequencing and Phylogenetic Analyses of the H12N8 Virus

Viral RNAs were extracted by using the TIANamp Virus RNA Kit (Tiangen), and standard reverse transcription PCR (RT-PCR) was performed with primers for AIVs (primer sequences available on request). The eight gene segments of the influenza virus were sequenced by Sangon Biotech Co., Ltd. (Shanghai, China). The nucleotide sequences were assembled using the SEQMAN program (DNASTAR, Madison, WI, USA). A Bayesian time-measured phylogenetic tree of the HA gene segment of the H12N8 virus was constructed using the BEAST software package (v1.10.4) with a relaxed uncorrelated lognormal molecular clock model [49]. Markov Chain Monte Carlo (MCMC) chains ran for 10 million generations. The Maximum Clade Credibility (MCC) trees were constructed by TreeAnnotator of the BEAST package and then visualized by the FigTree software (v1.4.4) as previously described [15,50]. The IQ-tree of the PhyloSuite platform was used to generate Maximum Likelihood trees, with nucleotide substitutions models selected using the ModelFinder algorithm [51,52]. The tree of NA was rooted to A/equine/Miami/1/1963(H3N8), and the trees of internal genes were rooted to A/equine/Prague/1956(H7N7). The sequences of the H12N8 virus were deposited in GISAID databases (https://www.gisaid.org, accessed on 29 August 2024) under accession numbers EPI3506790 to EPI3506797.

### 4.5. Virus Replication Curves in Cells

MDCK, A549, or DF1 cells were grown on 12-well plates and infected with the virus at an MOI of 0.01. The inoculum was removed after 1 h of incubation. After three washes with PBS, the cells were supplemented with DMEM containing tosylsulfonyl phenylalanyl chloromethyl ketone (TPCK)-treated trypsin and were incubated at 37 °C. The virus-containing cell supernatant was collected at 12 h, 24 h, 36 h, and 48 h p.i. and titrated in MDCK cells. The growth curve data shown are the results of three independent experiments.

### 4.6. Mouse Study

Six-week-old female BALB/c mice were intranasally inoculated with 10^6^ EID_50_ of the virus in a volume of 50 μL. Three mice were euthanized, and the brain, nasal turbinate, lung, spleen, and kidney were collected for virus titration in embryonated chicken eggs. Mice lungs were fixed in 10% formalin at 3 d p.i. and then stained with H&E for histological analysis. The remaining five mice were monitored daily for weight loss and survival over a 14-day period. Mice inoculated with PBS served as a control group to observe changes in body weight and any pathological effects.

### 4.7. Duck Study

To investigate the replication and pathogenicity of the virus in ducks, three 3-week-old SPF ducks were intranasally inoculated with 10^6^ EID_50_ of the virus in a volume of 200 μL. The organs, including the brain, trachea, lung, spleen, liver, pancreas, kidneys, intestine, rectum, and bursa of Fabricius, were collected at 3 d p.i. for virus titration in eggs. For the transmission study, three ducks were inoculated with 10^6^ EID_50_ of the virus in a volume of 200 μL. Three naive ducks of the contact group were placed into the same isolator at 24 h p.i. The oropharyngeal and cloacal swabs from both the inoculated and contact group ducks were collected at 2-day intervals, beginning at 2 d p.i. (one day post exposure). The viral titers of the swabs were titrated in eggs. Sera were collected on day 21 p.i., and the antibody titers were determined by using an HI assay.

### 4.8. Chicken Study

To investigate the replication and pathogenicity of the virus in chickens, three 3-week-old SPF chickens were intranasally inoculated with 10^6^ EID_50_ of the virus in a volume of 200 μL. The organs of three inoculated chickens, including brain, trachea, lung, spleen, liver, pancreas, kidneys, intestine, rectum, and bursa of Fabricius, were collected for virus titration in eggs at 3 d p.i. For the transmission study, three chickens were inoculated with 10^6^ EID_50_ of the virus in a volume of 200 μL. Three naive chickens of the contact group were placed into the same isolator at 24 h p.i. The oropharyngeal and cloacal swabs from both the inoculated and contact group chickens were collected at 2-day intervals, beginning at 2 d p.i. (one day post exposure). The viral titers of the swabs were titrated in embryonated chicken eggs. Sera were collected at 10, 15, and 21 d p.i., and the antibody titers were determined by using HI assay.

## Figures and Tables

**Figure 1 ijms-26-02740-f001:**
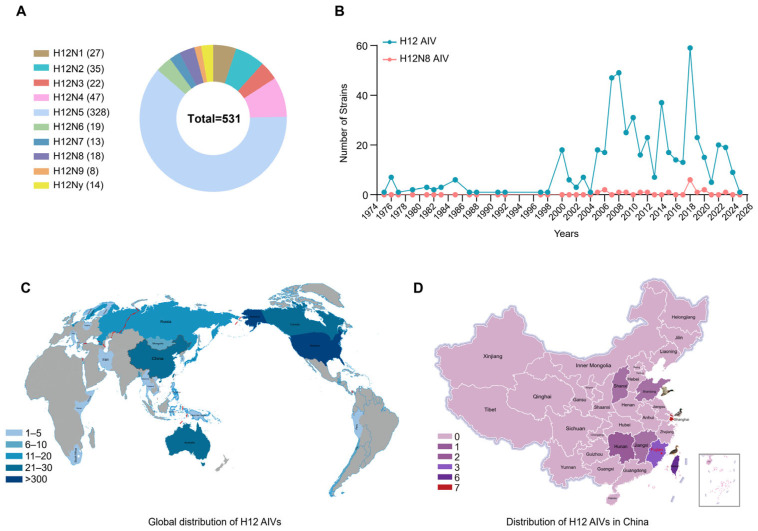
Global prevalence of H12 subtype viruses. (**A**) The HA and NA combinations of H12Ny strains in the database. (**B**) The isolation date of the H12 subtype viruses. (**C**) The global distribution of the H12 subtype viruses. (**D**) The distribution of the H12 subtype viruses in China. Sequences of the H12 subtype viruses were all retrieved in GISAID and GenBank databases to analyze the host, distribution, and isolation years of the viruses. All the public data utilized in this study from GISAID and GenBank were current as of 9 February 2025.

**Figure 2 ijms-26-02740-f002:**
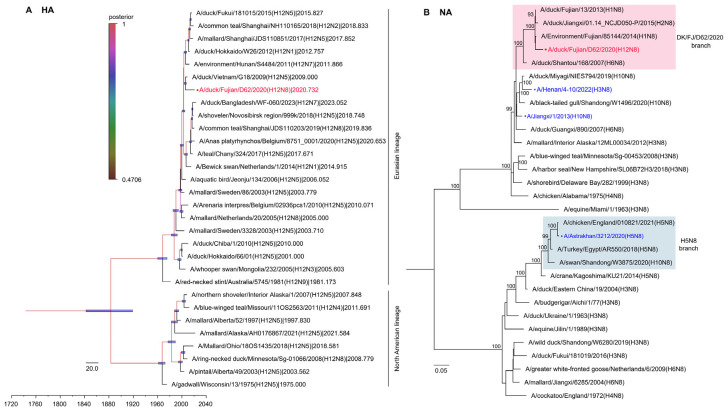
Phylogenetic analyses of HA and NA genes. (**A**) Maximum Clade Credibility (MCC) tree of the HA gene. Bayesian time-measured phylogenetic tree of HA was constructed with the BEAST software package (v1.10.4) and then visualized by using FigTree (v1.4.4). Branches are colored according to posterior probability, and the node bars indicate the 95% highest posterior density of the node height. (**B**) Phylogenetic tree of NA gene was constructed by employing the maximum-likelihood method with the IQ-TREE algorithm. The NA tree was rooted to A/equine/Miami/1/1963(H3N8). The H12N8 virus isolated in this study is shown in red, while human isolates were shown in blue. The regions of nucleotide sequence used for the phylogenetic analyses were as follows: HA, 29-1732, and NA, 19-1431.

**Figure 3 ijms-26-02740-f003:**
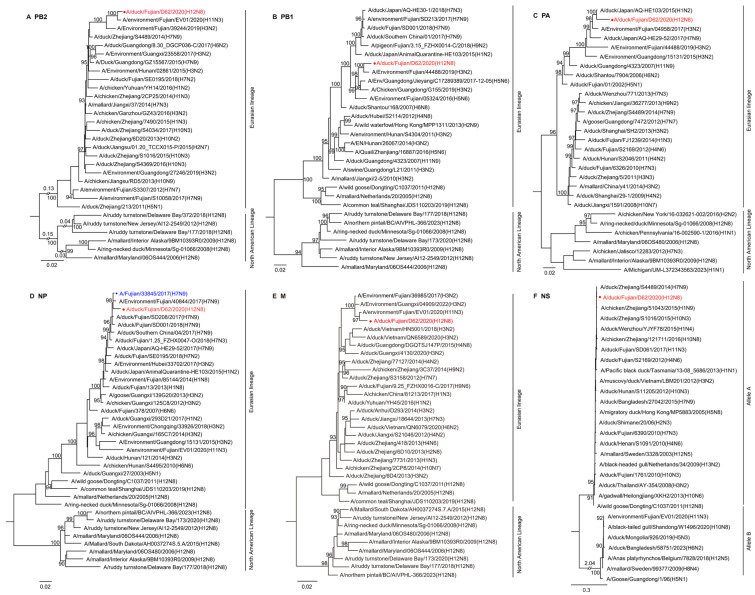
Phylogenetic analyses of the internal genes of H12N8 virus. Phylogenetic trees of the internal genes were generated by employing the maximum-likelihood method with the IQ-TREE algorithm. The regions of nucleotide sequence used for the phylogenetic analyses were as follows: PB2, 37-2307; PB1, 25-2298; PA, 25-2175; NP, 46-1542; M, 26-1007; and NS, 29-849. The trees of PB2 (**A**), PB1 (**B**), PA (**C**), NP (**D**), M (**E**), and NS (**F**) were rooted to A/equine/Prague/1/1956 (H7N7). The virus in red was used in this study, while the virus isolated from humans was shown in blue.

**Figure 4 ijms-26-02740-f004:**
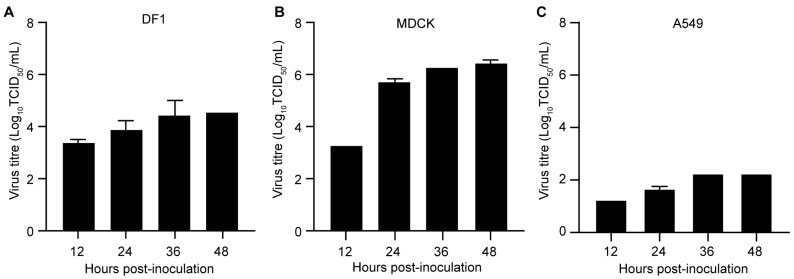
Replication of the H12N8 virus in avian and mammalian cells. (**A**) DF1 cells, (**B**) MDCK cells, or A549 cells (**C**) were infected with the H12N8 virus at an MOI of 0.01 in triplicate. Then, the supernatants were collected at the indicated times for titration in MDCK cells.

**Figure 5 ijms-26-02740-f005:**
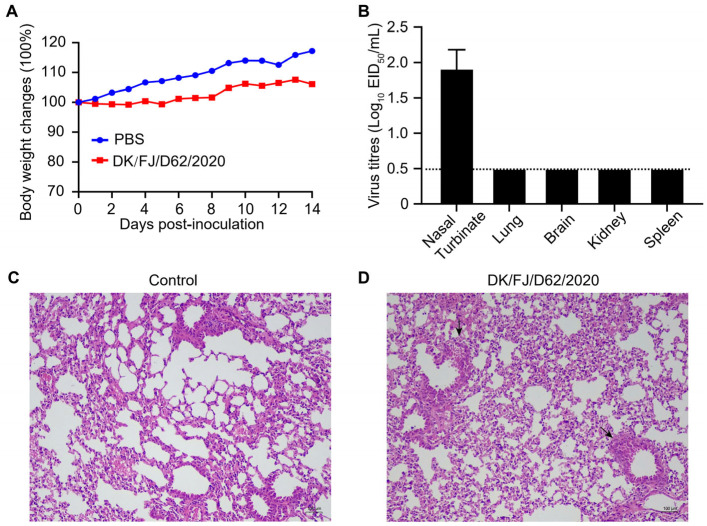
Replication and pathogenicity of the H12N8 viruses in BALB/c mice. (**A**) Body weight changes in mice after inoculation with 10^6^ EID_50_ of the DK/FJ/D62/2020 virus. (**B**) Virus replication titers in the indicated organs of mice after inoculation with 10^6^ EID_50_ of the DK/FJ/D62/2020 virus. Data shown are the mean virus titers (*n* = 3) ± standard deviation. The dashed line indicates the lower limit of virus detection. Hematoxylin and eosin (H&E) staining of the lungs of mice inoculated with PBS (**C**) or the H12N8 virus (**D**). Scale bars in (**C**,**D**) represent 100 μm.

**Figure 6 ijms-26-02740-f006:**
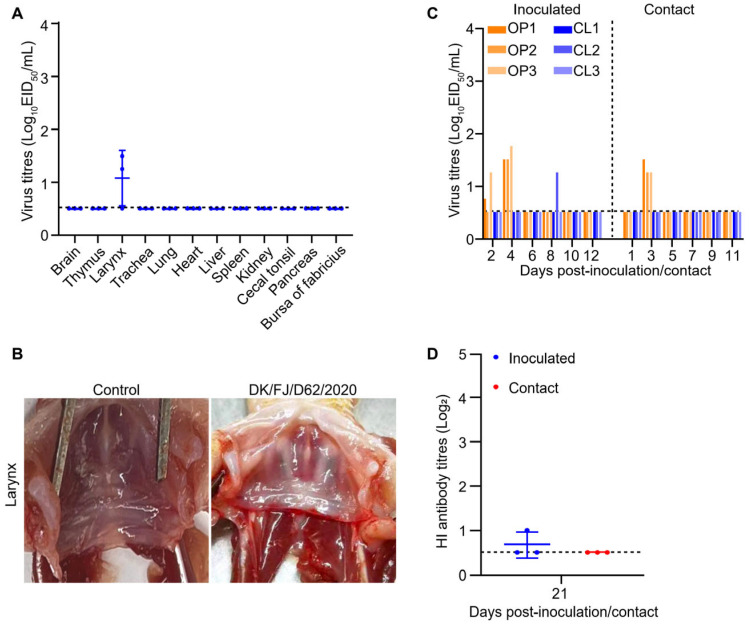
Replication and transmission of the H12N8 viruses in ducks. (**A**) SPF ducks were inoculated with the DK/FJ/D62/2020 virus. The indicated organs were collected at 3 d p.i. and viruses were titrated in embryonated chicken eggs. (**B**) Tissue damage at the laryngeal inlet of the ducks inoculated with PBS or the H12N8 virus. (**C**) Transmission of the H12N8 virus in ducks. Oropharyngeal and cloacal swabs were collected from the ducks at indicated times and the viruses were titrated in embryonated chicken eggs. (**D**) Serum from inoculated and contact ducks were collected at 21 d p.i. for HI titer detection. The dashed lines indicate the lower limit of virus detection in panels A and B and the lower limit of HI antibody titers detection in panel D. OP: oropharyngeal swab; CL: cloacal swab.

**Figure 7 ijms-26-02740-f007:**
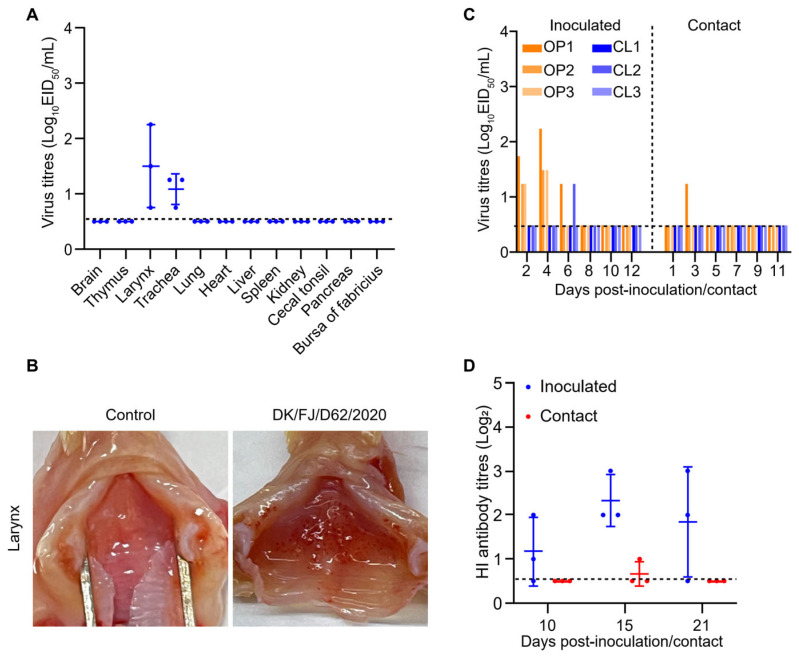
Replication and transmission of the H12N8 viruses in chickens. (**A**) SPF chickens were inoculated with the DK/FJ/D62/2020 virus. The indicated organs were collected at 3 d p.i. and viruses were titrated in embryonated chicken eggs. (**B**) Tissue damage at the laryngeal inlet of the chickens inoculated with PBS or H12N8 virus. (**C**) Transmission of the H12N8 virus in chickens. Oropharyngeal and cloacal swabs were collected from the chickens at indicated times and the viruses were titrated in embryonated chicken eggs. (**D**) Serum from inoculated and contact chickens was collected at 10, 15, and 21 d p.i. for HI titers detection. The dashed lines indicate the lower limit of virus detection in panels A and B and the lower limit of HI antibody titers detection in panel D. OP: oropharyngeal swab; CL: cloacal swab.

**Table 1 ijms-26-02740-t001:** Molecular characteristics of the DK/FJ/D62/2020 virus.

Protein	Key Amino Acid Mutations	Amino Acid of DK/FJ/D62/2020
HA	Q226L	Q
	G228S	G
NA	R292K	R
PB2	A588V	A
	G590S	G
	Q591K	Q
	E627K	E
	D701N	D
PB1	R207K	K
	I368V	I
	H436Y	Y
	M677T	T
PA	A515T	T
NP	V286A	A
	N319K	N
	M437T	T
M1	N30D	D
	T215A	A
M2	S31N	S
NS1	P42S	S

## Data Availability

The data supporting the conclusions of this article will be available from the corresponding author upon request.

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
