# Peer review of "Characteristics of the First Domestic Duck-Origin H12N8 Avian Influenza Virus in China"

_ijms, 2025, doi:10.3390/ijms26062740_

Round 1

Reviewer 1 Report

Comments and Suggestions for Authors

Authors described the isolation of H12N8 AIV from ducks from live poultry markets. They sequenced the eight genome segments and analyzed them by phylogenetic analysis demonstrating that HA gene displayed low similarity from dick H12N5 from neighboring country as Vietnam. Also, the other segments grouped with Eurasian lineage strains of AIVs.

They analyzed the viral replication of the isolated virus in mammal and chicken cells. The isolate could be replicated in both cells without prior adaptation showing the relevance in transmission of this subtype. They also analyzed the pathogenicity of H12N8 for mammals, chickens and ducks.  Their findings imply that ducks may serve as asymptomatic carriers of the H12N8 virus, indicating a potential for silent limited transmission within avian populations.

When AIV is replicated in poultry, they are likely to become highly pathogenic to chickens by undergoing extensive reassortments or genes mutations. So, they could have very important consequences such as H5/H7 HPAIV. Consequently, it’s important to control this special subtype that could cause severe damage over time to chickens. Thus, surveillance in poultry and wild birds is crucial for prevention.

The paper is well structured and presents an adequate analysis of the isolated AIV strain.  No further improvements are needed in the analysis.

Author Response

Response to Reviewer 1 Comments

1. Summary

  We sincerely thank the reviewers for their comprehensive and constructive feedback on our manuscript. We have carefully addressed each of the reviewers’ suggestions and believe that the revisions have significantly enhanced the quality of our manuscript. Below, we provide detailed point-by-point responses to the reviewers’ comments. The specific revisions are highlighted in track changes in the re-submitted files. We hope that these improvements now make our manuscript suitable for publication in IJMS.

2. Questions for General Evaluation

Reviewer’s Evaluation

Does the introduction provide sufficient background and include all relevant references?

Yes

Is the research design appropriate?

Yes

Are the methods adequately described?

Yes

Are the results clearly presented?

Yes

Are the conclusions supported by the results?

Yes

Response and Revisions:

We sincerely appreciate the reviewer's insightful comments and suggestions. We appreciate that you found the introduction sufficiently informative, with relevant references included, and the research design appropriate. We appreciate your affirmation that the conclusions are well supported by the data presented. Minor corresponding revisions, indicated in track changes, are included in the re-submitted files. We hope that the revisions will meet with your approval. We appreciate your guidance throughout this process.

3. Point-by-point response to Comments and Suggestions for Authors

Comments 1: Authors described the isolation of H12N8 AIV from ducks from live poultry markets. They sequenced the eight genome segments and analyzed them by phylogenetic analysis demonstrating that HA gene displayed low similarity from dick H12N5 from neighboring country as Vietnam. Also, the other segments grouped with Eurasian lineage strains of AIVs.

  They analyzed the viral replication of the isolated virus in mammal and chicken cells. The isolate could be replicated in both cells without prior adaptation showing the relevance in transmission of this subtype. They also analyzed the pathogenicity of H12N8 for mammals, chickens and ducks. Their findings imply that ducks may serve as asymptomatic carriers of the H12N8 virus, indicating a potential for silent limited transmission within avian populations.

  When AIV is replicated in poultry, they are likely to become highly pathogenic to chickens by undergoing extensive reassortments or genes mutations. So, they could have very important consequences such as H5/H7 HPAIV. Consequently, it’s important to control this special subtype that could cause severe damage over time to chickens. Thus, surveillance in poultry and wild birds is crucial for prevention.

  The paper is well structured and presents an adequate analysis of the isolated AIV strain.  No further improvements are needed in the analysis.

Response 1: We sincerely appreciate the reviewers’ insightful comments and suggestions. We are very grateful for your recognition of the structure and analysis of our paper. We are glad that our description of the isolation and genomic analysis of the H12N8 AIV from ducks in live poultry markets was clear and informative. The phylogenetic analysis, viral replication, and pathogenicity studies were indeed crucial aspects of our research. Your emphasis on the importance of surveillance in poultry and wild birds reinforces our study's significance.

4. Response to Comments on the Quality of English Language

Point 1: The English is fine and does not require any improvement.

Response 1: We sincerely appreciate the reviewers’ recognition of the language quality of our manuscript. We thoroughly checked the manuscript and made some minor revisions. These revisions are indicated in track changes in the re-submitted files.

Reviewer 2 Report

Comments and Suggestions for Authors

The authors have successfully characterized the domestic duck-origin H12N8 avian influenza virus (AIV) in China for the first time by isolating it from ducks in the live poultry market. This is a significant contribution to the understanding of emerging AIV subtypes, emphasizing the importance of active surveillance. The experimental design and methodology are well-executed, and the manuscript was generally well-written. The introduction, experiments, results, and discussion are appropriate and supported by experimental evidence and current literature. The results were very well supported by experimental evidence and discussion.  However, some areas of the manuscript, including the methods and results section and grammar, require minor improvements. Overall, the study was excellent, and the manuscript can be considered for publication following a few minor revisions.

Comments on the Quality of English Language

Need to be improved.

Author Response

Response to Reviewer 2 Comments

1. Summary

  We sincerely thank the reviewers for their meticulous review and constructive comments, which have been immensely helpful. We have carefully implemented the suggested revisions, and these changes have substantially enhanced the quality of our manuscript. Below, we provide detailed point-by-point responses to the reviewers' comments. The corresponding revisions are highlighted in track changes in the re-submitted files. We hope that the manuscript now meets the standards required for publication in IJMS.

2. Questions for General Evaluation

Reviewer’s Evaluation

Does the introduction provide sufficient background and include all relevant references?

Yes

Is the research design appropriate?

Yes

Are the methods adequately described?

Can be improved

Are the results clearly presented?

Can be improved

Are the conclusions supported by the results?

Yes

Response and Revisions:

  We sincerely appreciate that you found the introduction sufficiently informative, with relevant references included, and the research design appropriate. We've revised the Methods section, adding a detailed description of the virus isolation and identification procedures. We've also supplemented the Results section with more detailed information on virus isolation. The corresponding revisions, indicated in track changes, are included in the re-submitted files. We hope that the revisions will meet with your approval. We appreciate your guidance throughout this process.

3. Point-by-point response to Comments and Suggestions for Authors

Comments 1: The protocol for the identification of positive samples needs further classification. Specifically, why did the authors use HA activity rather than the gold standard real-time RT-PCR method to screen the samples for AIV?

Response 1: We sincerely appreciate the reviewer's insightful comments regarding the protocol for the identification of positive samples. We appreciate your suggestion to clarify the rationale behind our choice of methods. In our study, the samples were inoculated into 10-day-old embryonated chicken eggs for 48 h at 37℃. Then, the allantoic fluid was collected for testing HA activity as the initial screening method for AIV.

  Firstly, HA activity assays provide a quick and relatively inexpensive method for preliminary screening of huge samples. This is particularly useful in surveillance programs where high-throughput screening is necessary to identify potentially positive samples efficiently as described in our previous studies (PMID: 30269969 and PMID: 29151586 ).

  Secondly, while real-time RT-PCR is indeed the gold standard for detecting AIV due to its high sensitivity and specificity, HA activity assays serve as a valuable initial screening tool. Positive results from HA assays in our study are further confirmed using real-time RT-PCR or sequencing, thereby reducing the number of samples that need to undergo the more resource-intensive molecular testing.

  Lastly, HA activity assays directly measure the ability of the virus to agglutinate red blood cells, indicating the presence of infectious virus particles. This provides complementary information to nucleic acid-based detection methods like real-time RT-PCR, which can detect both infectious and non-infectious viral RNA.

  We acknowledge that real-time RT-PCR is the preferred method for definitive diagnosis and subtyping of AIV. In our study, we used HA activity as a preliminary screening method to identify samples that are likely to be positive, followed by confirmatory sequencing for accurate identification and subtyping of AIV in the positive samples. We have revised the manuscript to include a more detailed explanation of our rationale for using HA activity as an initial screening method in the protocol section.

Comments 2: In the results section, it would be beneficial to specify which year(s) out of the 1190 samples collected from 2019 to 2023 were positive for the AIV. The manuscript mentions that only one sample was positive, but more detail on the year of collection would be helpful for context. Furthermore, it is important to clarify the subsequent steps after the positive sample was identified. The manuscript should explicitly state that this sample was inoculated into SPF chicken embryo eggs for virus isolation. Following virus isolation, the eight segments of the influenza virus should be amplified using AlV-specific primers by RT-PCR, and the amplified nucleic acid should be sent for Sanger sequencing, as described in the methodology. Clearly outlining these steps will give the reader a more comprehensive understanding of the experimental process.

Response 2: We sincerely appreciate the reviewer's insightful comments and suggestions. We have comprehensively elaborated on the annual sample collection and provided an in-depth description of the virus isolation process in lines 99-105. In the results section, we have also thoroughly detailed the outcomes of the virus isolation and identification in lines 105-109. The revised is shown as “We collected 25, 209, 112, 131, and 713 samples from 2019 to 2023 annually. The samples were inoculated into 10-day-old embryonated chicken eggs for virus isolation. Subsequnently, the positive sample was amplified in Specific pathogen free (SPF) chicken embryos eggs, and their HA and NA subtype were identified using sanger sequencing. Virus isolation yielded 116 AIVs, with annual distribution as follows: 26 isolates in 2020, 17 in 2021, 5 in 2022, and 68 in 2023. The subtype distribution of these viruses is H3 (30.17%, 35/116), H6 (27.59%, 32/116), H4 (16.38%, 19/116), and H9 (12.93%, 15/116), H10 (5.17%, 6/116), H1 (4.31%, 5/116), H11 (2.59%, 3/116), and H12 (0.86%, 1/116).

Comments 3: The phylogenetic trees are difficult to interpret and should be improved for better clarity and illustration. Enhancing the readability of these figures will help the manuscript's presentation.

Response 3: We sincerely appreciate the reviewer's insightful comments and suggestions. We fully agree that the readability of these figures is crucial for the presentation of our manuscript. To enhance the clarity and interpretability of phylogenetic trees, we removed some redundant sequences and reconstructed the phylogenetic trees for HA and NA in Figure 2.

  Specifically, we have made the following improvements. Firstly, we have adjusted the tree layout to ensure that branches are more evenly spaced and that the overall structure is more intuitive. This includes modifying the branch angles and lengths to better reflect evolutionary distances while avoiding overcrowding. Secondly, we have added clearer labels for each branch and node, including virus strain names, subtypes, and relevant metadata. Additionally, we have included branch support values to indicate the reliability of the inferred relationships. Lastly, to improve visual distinction, we have introduced color coding to differentiate among various subtypes or groups of viruses. The H12N8 virus isolated in this study is shown in red, while the human isolates are shown in blue.

   We hope that these revisions have significantly improved the readability and interpretability of the phylogenetic trees. The updated figures are included in the revised manuscript, and we have highlighted the changes in track changes for your reference. Thank you again for your constructive feedback. We hope these improvements meet your expectations and enhance the overall quality of our manuscript.

Comments 4: Table 1: There is an inconsistency between the explanation in the results section (lines 104-107) and the data shown in Table 1. The text mentions: "The amino acids at the receptor binding site in the HA protein are Q226 and G228 (H3 numbering), which indicates that this virus may preferentially bind to avian-type receptors (a-2,3 sialic acid) (Table 1)." However, in Table 1, the mutations listed under the second column titled "Amino acid mutations" are Q226L and G228S. This creates confusion as the amino acid positions are indicated differently, and the mutations presented do not align with the information described in the results section. Additionally, the column title "Amino acid mutations" does not clearly correlate with the text "Cleavage site" mentioned in the description of Table 1, Please revise Table 1 and Clarify the column titles and description.

Response 4: We sincerely appreciate the reviewer's insightful comments and suggestions. We have revised the column titles and descriptions in Table 1 to ensure clarity and consistency. Column 2 listed in Table 1 are key amino acid mutations that influence virus receptor binding property, pathogenicity, or transmissibility reported by researchers. Column 3 are the amino acids of DK/FJ/D62/2020(H12N8) virus. Thus, amino acids at 226 and 228 sites are Q and G, respectively, corresponding to the property that the virus may bind to avian-type receptors (a-2,3 sialic acid). We have revised this section to accurately reflect the data presented in Table 1. Since we have clearly demonstrated the cleavage site of the H12N8 virus in line 112 in the test, we deleted the redundant information in Table 1. The column titled "Key amino acid mutations" now clearly correlates with the text and accurately reflects the data. We hope that these revisions have resolved the inconsistencies and improved the clarity of Table 1. The changes are highlighted in the revised manuscript for your reference.

Comments 5: Only a few (but certainly not all) examples are given in red font for changes to illustrate my points regarding the grammar.

Line:9, contribute to contributed

Line:16, bird origin to bird-originated

Line:29-30, Orthomyxoviridae family, with its genome consisting of eight negative-sense, single-stranded RNA segments

Line: 38: Wild aquatic birds are add the primary

Line: 45: for add the recombination

Line: 46, 47: H3N8, H7N9, and H10N8, with their genes derived from duck-origin viruses,

Line: 54: diverse add origins

Line: 65: add a live

Line: 65: add a reassortant

Line: 70: ducks

Line: 73: add the poultry

Response 5:  We sincerely appreciate the reviewers’ insightful comments and suggestions. The revised grammar section listed above are shown in line 9, line 16, line 29-30, line 37, line 44-47, line 54, line 65, line 68, line 70, and line 73 in the revised manuscript. We have conducted a thorough review of the manuscript, making corrections to meet the rigorous standards of academic publishing and to improve readability for our audience. All the changes have been highlighted with track changes in the revised manuscript.

4. Response to Comments on the Quality of English Language

Point 1: The English could be improved to more clearly express the research.

Response 1: We sincerely appreciate the reviewers’ insightful comments and suggestions. We recognize the importance of clear and effective communication in scientific research and are grateful for your suggestions to enhance the clarity of our work. We have carefully revised the English in our manuscript to ensure that our research is presented as clearly and accurately as possible. We have paid particular attention to the sections you highlighted and have made the necessary adjustments to improve the overall readability and precision of our writing. The changes have been highlighted with track changes in the revised manuscript.